# Genome Size Variation in *Sesamum indicum* L. Germplasm from Niger

**DOI:** 10.3390/genes15060711

**Published:** 2024-05-29

**Authors:** Najat Takvorian, Hamissou Zangui, Abdel Kader Naino Jika, Aïda Alouane, Sonja Siljak-Yakovlev

**Affiliations:** 1Université Paris-Saclay, CNRS, AgroParisTech, Ecologie Systématique Evolution, 91190 Gif-sur-Yvette, France; aida.alouane@sorbonne-universite.fr; 2Sorbonne Université, UFR Sciences de la Vie, UFR927, 4 Place Jussieu, F-75005 Paris Cedex 05, France; 3Department of Plant Production, Abdou Moumouni University, BP-10960 Niamey, Niger; zanguiagro@gmail.com (H.Z.); kaderjika@gmail.com (A.K.N.J.)

**Keywords:** germplasm, flow cytometry, intraspecific variation, sesame, statistical analysis, total nuclear DNA content, 2C DNA value

## Abstract

*Sesamum indicum* L. (Pedaliaceae) is one of the most economically important oil crops in the world, thanks to the high oil content of its seeds and its nutritional value. It is cultivated all over the world, mainly in Asia and Africa. Well adapted to arid environments, sesame offers a good opportunity as an alternative subsistence crop for farmers in Africa, particularly Niger, to cope with climate change. For the first time, the variation in genome size among 75 accessions of the Nigerien germplasm was studied. The sample was collected throughout Niger, revealing various morphological, biochemical and phenological traits. For comparison, an additional accession from Thailand was evaluated as an available Asian representative. In the Niger sample, the 2C DNA value ranged from 0.77 to 1 pg (753 to 978 Mbp), with an average of 0.85 ± 0.037 pg (831 Mbp). Statistical analysis showed a significant difference in 2C DNA values among 58 pairs of Niger accessions (*p*-value < 0.05). This significant variation indicates the likely genetic diversity of sesame germplasm, offering valuable insights into its possible potential for climate-resilient agriculture. Our results therefore raise a fundamental question: is intraspecific variability in the genome size of Nigerien sesame correlated with specific morphological and physiological traits?

## 1. Introduction

The genus *Sesamum* comprises between 20 and 30 accepted species [1,2] including sesame, a widely cultivated species that offers many varieties with different seed colors. Sesame (*S. indicum* L., family Pedaliaceae) is one of the oldest oil crops (~3000 BCE) [3]. It is an annual herbaceous diploid plant (2n = 26) commonly known as the “Queen of oilseeds”, highly appreciated for its high-quality oil content (50–60%), natural antioxidants, protein content (18–28%) [4,5] as well as many other bioactive components [5]. Edible sesame seeds are therefore widely used in the food, pharmaceutical and cosmetic industries [6,7,8].

Sesame is mainly cultivated in the tropical and subtropical areas of the world [4,9]. Numerous wild relative species have been found in Africa and a smaller number in India [1]. Sesame cultivation is well adapted to arid environments and offers Africa in general and for Niger in particular, a good economical and national opportunity in the changing climatic conditions that have become unsuitable for other crops. World sesame production reached 6,741,479.41 tons in 2022 [10]. African production accounts for 59.3% of this world production (4,000,119 tons) with Niger contributing with 104,088.04 tons [10] representing a 20% increase in Niger’s production in 2021. 

The African germplasm remains still poorly characterized compared to the Asian germplasm, which has benefited from powerful genomic and genetic investigations [9,11,12,13,14,15,16,17,18,19]. Dossa et al. [20] carried out an analysis of the genetic diversity and population structure of sesame accessions originating mainly from west Africa and Asia and showed that the African accessions have lower genetic diversity than the Asian accession. However, in our previous work, preliminary analyses using AFLP markers revealed high genetic diversity in the Nigerien sesame germplasm [21]. In addition to these genetic studies, sesame has been extensively characterized morphologically and biochemically [1,5,22,23,24]. The accessions of the Nigerien sesame show great agro-morphological and biochemical diversity [22,23,24]. However, the data on the genome size (GS) are too sparse. For the Asian accession, genome sequencing allows to estimate the GS [9,13,15,16,17,19], but only two reports concern the total nuclear DNA amount (2C DNA) obtained via flow cytometry [15,25]. 

The genome size or 2C DNA is one of the most fundamental biological traits [26]. Swift [27] was the first to propose the term “C-value” as the DNA content of the unreplicated gametic chromosome set of an individual, considered to be invariable (C for constant). Since then, the intraspecific variation in the GS has often been detected [25,28,29]. The GS is a highly relevant trait that correlates with many biotic and abiotic characteristics [26,30]. Therefore, the knowledge of the DNA amount is currently an invaluable feature in a number of disciplines, including ecology and phytogeography [31,32,33], systematics and evolution [34,35,36,37], biodiversity screening [29,38,39,40] and also biotechnology and agricultural sciences [41,42,43]. 

Flow cytometry is currently the most widely used method for assessing the nuclear DNA amounts [44,45] because it is the most accurate and easiest to use. Our study is the first analysis of the GS via flow cytometry carried out on Nigerien sesame accessions. The aim was to explore the potential intraspecific GS variation in this germplasm, which presents variable morphological, biochemical and phenological characteristics [22,23,24].

## 2. Materials and Methods

### 2.1. Plant Materials

The plant material studied included 75 of the 140 accessions of *S. indicum* collected from 2015 to 2016, in 44 localities from 6 regions of Niger (Figure 1, Table 1). These regions belong to the agroecological zones where sesame production is the most important in Niger. The seventy-five accessions in the Niger germplasm were selected on the basis of their membership of the previously identified genetic and/or agro-morphological groups [21,22] and their geographical distribution (Figure 1, Table 1). Most of them had already been biochemically characterized [23]. A Tai sesame accession (STh) purchased on the market was included in the studied sample. The seeds of each accession were germinated in Petri dishes, and cotyledons were used for the GS evaluation.

### 2.2. Genome Size Assessment Using Flow Cytometry

In order to isolate the nuclei for cytometric analysis, the cotyledons from the germinated seedlings of *S. indicum* and the leaves from the internal standard (tomato, *Solanum lycopersicum* L. ‘Montfavet 63-5’, 2C = 1.99 pg [46]) were co-chopped using a razor blade in a Petri dish containing 1 mL of cold Gif nuclear isolating buffer GNB: 45 mM MgCl_2_, 30 mM Sodium-Citrate and 60 mM MOPS acid pH 7.0, 1% PVP 10.000, RNAse (2.5 U/mL) and 10 mM sodium metabisulfite (S_2_O5.Na_2_), which is a reducing agent that is less toxic than β-mercaptoethanol [44]. The nuclei suspension was filtered through a nylon mesh (Partec-CellTrics, pore size of 30 μm) and stained using a specific DNA fluorochrome intercalating dye propidium iodide (stock 1 mg/mL, Sigma-Aldrich, F-38297 Saint-Quentin-Fallavier Cedex, France) to a final concentration of 50 μg/mL, and kept for 5 min at 4 °C.

At least five individuals from each accession were analyzed for their average GS. The total 2C DNA content of at least 2000 stained nuclei was determined for each sample using a CytoFLEX S (Beckman Coulter, F-93420 Villepinte, France—Life Science United States (excitation 488 nm (50 mW) or 561 nm (30 mW); emission through a 690/50 or 610/20 nm band-pass filter, respectively, to lasers). Fluorescence histograms were analyzed using Kaluza software version 2.1 (Beckman Coulter) for each sample and internal standard. The nuclear DNA content was estimated using the linear relationship between the fluorescent signals from the stained nuclei of the *S. indicum* sample and the internal standard obtained via the following formula:2C DNA (pg)/nucleus = (Sample 2C peak mean/Standard 2C peak mean) × Standard 2C DNA (pg)

The mean of the 1Cx value (monoploid genome size) was calculated taking into account that 1 pg = 978 Mbp [47]. 

### 2.3. Statistical Analyses

Statistical analyses were carried out on all the data which were entered into Excel and analyzed using the R software version 4.3.2 developed by the Core Team of 2023 [48]. Descriptive statistics (mean, minimum, maximum, standard error, coefficient of variation) were first calculated for each accession (Appendix A). 

The non-parametric Kruskall–Wallis test was used to check for any differences in the GS among the 76 accessions of *S. indicum*. Due to the small sample size, 10,000 random permutations of individuals were also performed to calculate each p-value, ensuring the significance of the Kruskall–Wallis test. In the event of a significant effect on the sample, post-hoc non-parametric tests were used for the pairwise comparisons of sample medians using Dunn’s test [49]. Adjusted p-values for multiple testing were calculated using Holm’s method [50]. Spearman correlation coefficients were also used to test the relationship between the GS and various variables: branching (number of primary lateral branches at physiological maturity), fatty acid content (percentage of total fatty acids per unit dry matter), flowering time (date on which 50% of plants in a plot flower), height (from the crown to the top of the plant, measured at harvest in cm), latitude and longitude (Table 1), seed maturity (date at which all plants are fully mature), tegument color of mature seeds after harvest and yield (weight of total seeds harvested from a plant in g/plant). The data on the genetic groups (based on AFLP markers) and agro-morphological groups (determined by using classification methods based on the agro-morphological data from the experimental trials) were obtained from previous studies [21,22,23,24] that used the same seed lots as in the present work. The chi-squared test was performed to examine the relationship between the genome size and flowering time.

## 3. Results

### 3.1. Genome Size of Nigerien and Tai’s Accessions 

The 2C DNA values obtained for 76 *S. indicum* accessions are presented in Table 2 and Appendix A. Individual GSs ranged from 2C = 0.77 pg for S86, S91, S96 and S104 to 2C = 1.00 pg in accession S132 for the entire Niger panel (Table 2 and Appendix A). The mean GS ranged from 2C = 0.78 ± 0.008 pg in accession S96 to 2C = 0.95 ± 0.031 pg in accession S132 (Figure 2A,B, and Appendix A; Table 2 and Appendix A). The mean 1Cx value (monoploid genome size) of the estimates of the Nigerien accessions therefore ranged from 382 Mbp for accession S96 to 464 Mbp for accession S132 (Table 2). The GS of the Tai STh accession was the smallest (2C = 0.73 ± 0.01 pg; 1 Cx = 356 Mbp) compared to the GS of the Nigerien accessions (Table 2 and Appendix A) and also showed the smallest individual value (2C = 0.72 pg; 352 Mbp) (Figure 2C and Appendix A).

### 3.2. Analysis of the Genome Size Variation among the Nigerien Sesame Accessions

All the accessions studied showed a fairly wide range of 2C values and an overall coefficient of variation of 4.4% to 4.8% depending on whether or not the STh GS was included (Appendix A). When estimated within the genetic (Gr) and agro-morphological (AgroM) groups (Table 2 and [21,22]), the coefficient of variation was 3.7%, 4.8% and 4.3% in Gr1, Gr2 and Gr3, respectively, and 4.5%, 4.4% and 4.2% for AgroM1, AgroM2 and AgroM3, respectively (Appendix A). 

Kruskall–Wallis tests were performed on all the scored GS data from the accessions studied and among or within the Nigerien genetic and agro-morphological groups (Table 2 and Table 3). The analysis undertaken with all the GS scored data including or excluding the STh accession provided strong evidence of a highly significant difference (*p* < 0.001) between at least one pair of GS sesame accessions under study (Table 3). Dunn’s pairwise test was then performed among the Niger sesame accessions, and the results showed a significant difference (*p* < 0.05) among 58 pairs of the GS sesame accessions (Appendix A). When the GS of the STh accession was included in Dunn’s test, 68 pairs of accessions showed a significant variation in the GS (*p* < 0.05), with 19 of them showing a significant difference in the GS compared with the Tai accession (*p* < 0.05). 

The Kruskall–Wallis tests revealed a significant GS variation (*p* < 0.001 and *p* < 0.01) within and among the genetic and agro-morphological groups (Figure 3, Table 2 and Table 3). Dunn’s pairwise test including or excluding the STh accession (Appendix A) revealed a significant GS variation between the Gr2 and Gr3 genetic groups (*p* < 0.01), the AgroM1 and AgroM2 groups (*p* < 0.05) and the AgroM2 and AgroM3 groups (*p* < 0.01). When STh was excluded from the pairwise test, the GS variation between AgroM2 and AgroM3 groups was highly significant (*p* = 0.001). All the genetic and agro-morphological groups displayed a highly significant GS variation (*p* = 0.00) compared to the Tai STh accession used as an independent group (Figure 3, Appendix A).

### 3.3. Correlation between the Genome Size and Flowering Time

Among all the variables tested, a significant correlation for the Nigerien GS accessions was observed only between the GS and flowering time (Table 4). Although the Spearman’s correlation coefficient was moderate (r = −0.27), it was still statistically significant (*p* = 0.02).

The linear regression model showed a statistically significant (*p* < 0.05) but relatively weak association between the GS and flowering time (Figure 4, Table 5). In fact, only 4.5% of the variation in flowering time can be explained by the GS values. No correlation was found between the GS and flowering time when the analysis was carried out within the genetic or agro-morphological groups (Table 5). 

## 4. Discussion

Previous information on the GS of *S. indicum* was mainly based on the whole genome sequencing of Asian landraces [9,13,15,19]. Although genome sequencing provides valuable genomic information, the estimates of the GS derived from this method often underestimate the true total amount of DNA due to the lack of data on repetitive DNA, which is difficult to assemble using short-read sequencing technologies [51]. The improved assembly and annotation of the sesame genome using long-read sequencing and Hi-C technologies have resulted in an updated sequence [9], which is still lower than the GS expected by flow cytometry [15] or *k-mer* analysis [9]. Flow cytometry, particularly when using internal standards with known and stable GS, is the most reliable approach [44,45], especially for analyzing the intraspecific variation in the GS. Despite the importance of accurate GS determination, flow cytometry has only been used in two studies to assess the GS in *S. indicum* [15,25]. In our study, flow cytometry, used on a large sample of sesame accessions from Niger, provided evidence of an intraspecific variation in the GS in *S. indicum*. The Niger sesame accessions have previously been structured into three groups on the basis of agro-morphological traits or genetic criteria [21,22,24]. Statistical analyses showed that the GS varied among and within the genetic and agro-morphological groups (Figure 3, Table 3). Pairwise comparison showed that the GS varied significantly between certain genetic and agro-morphological groups (Appendix A). The lack of information on growing conditions does not allow to explain these GS variations at present.

The Niger sesame accessions had a higher average GS (1C = 0.43 pg; 420 Mbp) than the Chinese sesame cultivar ‘Zhongzhi No. 13’ for which the 1C value, estimated by flow cytometry, was 0.34 pg (337 Mbp) [15]. This difference in C-values could be due to the use of different cytometers and internal standards. The GS of STh, a Tai sesame accession, and the only Asian representative available, showed a significantly smaller GS (2C = 0.73 ± 0.01 pg; 1C = 356 Mbp) than our panel of sesame accessions. It would be interesting to extend this study to a larger sample to check whether the GS of the Asian sesame is generally smaller than the African sesame.

The Plant DNA C-values Database [25] provides the GS estimates for ten species of the genus *Sesamum*, including *S. indicum* with 2C = 1.91 pg (1C = 934 Mbp). This 2C value is approximately 2- to 2.45-fold higher than those of the Niger accessions. This value could be due to an error in the GS estimation or to the consequence of the polyploidy or amplification of repetitive DNA sequences, two mechanisms contributing to the increase in the GS [30,52]. However, in our Niger sesame panel, no cases of polyploidy were observed. Repetitive DNA is estimated to represent 52.81% of the updated sesame genome assembly [9] with the transposable elements (TEs) being the most abundant repetitive elements (27.42%). With the divergence rates estimated at less than 20% for the TEs and other repetitive elements, it has been suggested that the significant recent activity has led to their progressive accumulation in the sesame genome [9]. With the sesame genome sequence available [9], it would be relevant to isolate different types of repeat sequences (such as the TEs and satellite DNA) and use them as probes for in situ hybridization (FISH) on chromosomes. By comparing the FISH profiles of different sesame accessions, it would be possible to assess their contribution to the GS variation.

The variation in the GS among species is widely documented, particularly in plants [30], and its impacts play a key role in plant biodiversity, ecology and evolution [30,39,52,53,54,55,56]. There are also several reports on the relationship between the intraspecific variation in the GS and the phenotypic, phenological or ecological factors [28,31,34,57,58,59,60]. Indeed, the inter- and intra-specific variation in the GS may have ecological and evolutionary significance, as it has been correlated with the various phenotypic traits in plants such as cell size [57], seed mass [60], flower size [61], leaf size and metabolic rates [62], growth rate [63] and flowering time [59,64,65]. 

An investigation of the correlation between the variation in the GS of the Niger sesame and different traits revealed only an inverse relationship between the GS and flowering time, as shown by the moderate negative correlation between the two variables (Figure 4, Table 4 and Table 5), mainly attributed to the variation in the GS among the accessions. The moderate correlation revealed in the present study reflects the weak effect of the GS on the flowering time in the Niger sesame, as has been suggested for maize [66]. Plants with small genomes have a shorter cell cycle, rapid growing periods [57,67,68] and seemed to be favored in environments with harsh conditions such as low and high temperatures or nutrient deficiencies [33,69,70].

Despite the biochemical and morphological diversity reported among the Niger sesame accessions [22,23,24], no correlation was found between the GS and their representative variables, nor with the geographical variables (Table 4). However, in other cases, such as *Geranium macrorrhizum*, the chemical composition of the essential oils was correlated with polyploidy and consequently with the GS [58]. In *Silene latifolia*, the flower size was closely linked to the 2C DNA values [61]. In future studies, we will look for any correlation between the GS variation and the morphological and phytochemical characteristics of the African sesame.

## 5. Conclusions

The present work focused on the variation in the nuclear DNA content of *S. indicum*. In our panel of 75 accessions from the Niger germplasm and one Asian accession, several important results can be highlighted. This is the first time that the GS of *S. indicum* has been accurately determined from a large sample. The intraspecific variability in the GS was observed among germplasm accessions and a significant difference in the nuclear DNA between the Nigerien and Asian representatives was detected. A moderate but significant negative correlation was also observed between the GS and flowering time.

## Figures and Tables

**Figure 1 genes-15-00711-f001:**
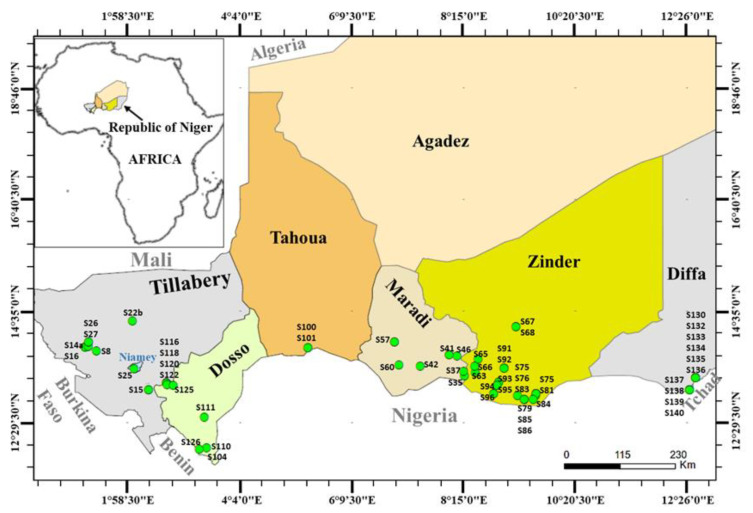
Spatial distribution of Nigerien sesame studied accessions. The map shows the different states of Niger and green circles indicate collection sites. Niger sesame accessions were numbered with S for sesame followed by a number.

**Figure 2 genes-15-00711-f002:**
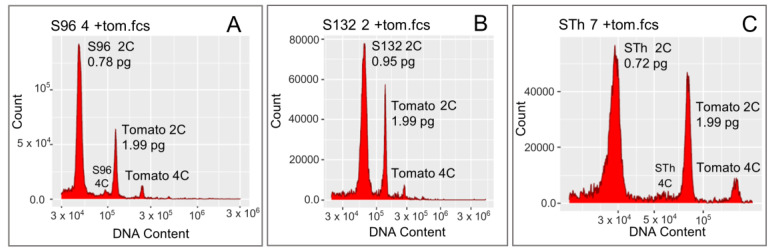
Flow cytometry histograms showing the fluorescence intensity peaks of samples and the internal standard: (**A**) the smallest GS (S96), (**B**) the largest GS (S132) among the Niger accessions and (**C**) the smallest GS Tai accession (STh).

**Figure 3 genes-15-00711-f003:**
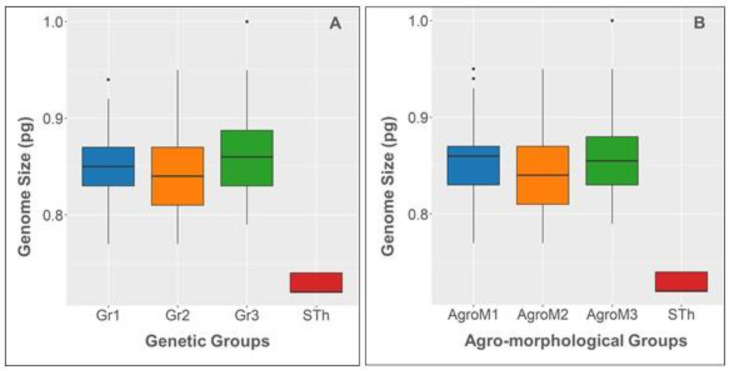
Box plots of the mean 2C values of genetic (**A**) and agro-morphological groups (**B**) of the Niger sesame. The Tai STh accession is included as an independent group. Horizontal lines denote median values. Boxes below and above the median line indicate the first and the third quartile. Extended vertical lines indicate extreme values. Dots indicate outliers.

**Figure 4 genes-15-00711-f004:**
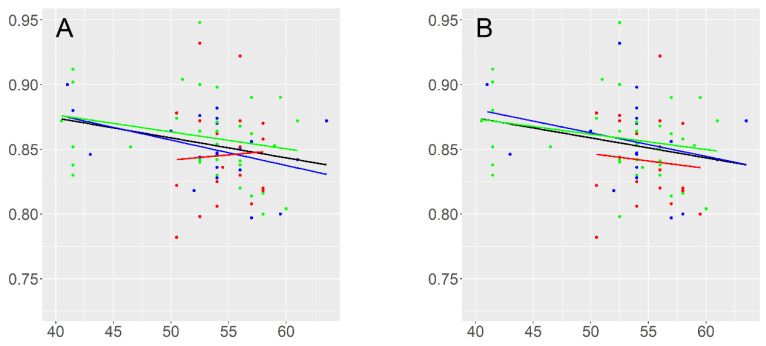
Correlation between the genome size and flowering time between and within the genetic (**A**) and agro-morphological (**B**) groups. The black line shows the linear regression from all GS data. The blue, red and green lines represent the linear regression within Gr1 or AgroM1, Gr2 or AgroM2 and Gr3 or AgrM3, respectively.

**Table 1 genes-15-00711-t001:** Origin of the studied accessions with GPS coordinates.

Accessions		Locality	GPS Coordinates	
Name	Tegument Color		Latitude	Longitude
S1	brown	Koulbaga	13°55′00.7″	1°14′39.2″
S2	brown	Koulbaga	14°24′59.9″	3°19′26.8″
S3	beige	Koulbaga	14°24′59.9″	3°19′26.8″
S4	beige	Koulbaga	14°24′59.9″	3°19′26.8″
S7	gray	Guéro kiraï	13°54′48.9″	1°14′19.2″
S8	beige	Tchoumbo	13°54′54.0″	1°14′20.3″
S11	white	Garbey kourou	13°74′22.7″	1°61′26.4″
S12	beige	Garbey kourou	13°74′22.7″	1°61′26.4″
S13	gray	Garbey kourou	13°74′22.7″	1°61′26.4″
S14a	beige	Djabou	14°00′04.6″	1°14′31.9″
S15	brown	Say	13°09′27.5″	2°22′25.6″
S16	brown	Djabou	14°00′04.6″	1°14′31.9″
S17	brown	Say	13°09′27.5″	2°22′25.6″
S22b	beige	Kossa	14°23′24.2″	2°03′38.4″
S25	beige	Sona	13°49′97.4″	2°09′36.6″
S26	beige	Lossa	13°92′11.2″	1°57′55.4″
S27	brown	Lossa	13°92′11.2″	1°57′55.4″
S28	brown	Ingonga	13°34′08.3″	2°31’67.4″
S35	white	Hawan Dawaki	13°55’70.2″	2°90’45.8″
S37	beige	Korgom	13°45’25.2″	8°25’60.5″
S41	white	Aguié	13°75’58.7″	7°98’86.4″
S42	white	Tchadoua	13°54’78.4″	7°44’77.7″
S44	white	Dan Jikaou	13°31’22.2″	7°54’41.0″
S46	white	Tessaoua	13°73’87.5″	8°13’79.0″
S50	white	Kibya Ga Kougou	13°73’71.7″	6°98’11.7″
S52	beige	Kibya Ga Kougou	13°73’71.7″	6°98’11.7″
S53	white	Guidan Tanko	13°48’37.6″	2°59’52.3″
S54	white	Sabon Machi	13°87’58.1″	6°98’04.1″
S57	beige	Jaja	14°00′11.6″	6°96′45.9″
S60	brunette	Tibiri	13°57′07.3″	7°05′06.4″
S63	white	Matameye	13°42′57.3″	8°48′01.0″
S65	beige	Takiéta	13°67′99.4″	8°52′95.9″
S66	brunette	Kantché	13°42′57.3″	8°48′01.1″
S67	beige	Dan Bouta	13°76′60.2″	8°74′30.8″
S68	gray	Dan Bouta	13°76′60.2″	8°74′30.8″
S71	beige	Ga Allah	12°99′63.1″	9°26′86.8″
S72	beige	Dan Koublé	13°01′53.2″	8°49′13.4″
S75	beige	Malawa	13°03′12.5″	9°61′34.1″
S76	brunette	Malawa	13°03′12.5″	9°61′34.1″
S79	brunette	Hayaniya	12°92′55.3″	9°39′30.1″
S81	beige	Malawa	13°03′12.5″	9°61′34.1″
S82	white	Katta Kara	12°99′13.3″	9°60′49.1″
S83	beige	Malawa	13°03′12.5″	9°61′34.1″
S84	beige	Dan Marké	12°92′12.2″	9°56′08.7″
S85	beige	Hayaniya	12°92′55.3″	9°39′30.1″
S86	white	Hayaniya	12°92′55.3″	9°39′30.1″
S91	brown	Dogo	13°50′21.2″	9°01′59.6″
S92	beige	Dogo	13°50′21.2″	9°01′59.6″
S93	white	Kaba	13°24′45.1″	8°91′70.9″
S94	beige	Bandé	13°18′21.2″	8°88′64.4″
S95	brunette	Kaba	13°24′45.1″	8°91′70.9″
S96	brunette	Bandé	13°18′21.2″	8°88′64.4″
S99	beige	Dogo	13°50′21.2″	9°01′59.6″
S100	beige	Tsarnaoua	13°89′23.0″	5°34′43.4″
S101	brown	Tsarnaoua	13°89′23.0″	5°34′43.4″
S104	brunette	Tara	11°89′96.6″	3°33′50.9′
S110	brown	Gaya	12°00′57.6″	3°27′12.3″
S111	beige	Guéza	12°35′29.9″	3°24′25.5″
S116	beige	Kodo	13°14′04.4″	2°41′59.4″
S118	brunette	Kodo	13°14′04.4″	2°41′59.4″
S120	beige	Kodo	13°14′04.4″	2°41′59.4″
S122	black	Kodo	13°14′04.4″	2°41′59.4″
S123	black	Dara Salam	13°19′33.0″	12°36′0.30″
S125	white	Madina	13°10′45.8″	2°49′40.1″
S126	white	Tanda	11°59′33.5″	3°18′40.2″
S130	beige	Diffa	13°19′33.0″	12°36′00.3″
S132	white	Diffa	13°19′33.0″	12°36′00.3″
S133	beige	Diffa	13°19′33.0″	12°36′00.3″
S134	beige	Diffa	13°19′33.0″	12°36′00.3″
S135	beige	Gandaré	13°19′33.0″	12°36′00.3″
S136	beige	Gandaré	13°19′33.0″	12°36′00.3″
S137	beige	Geida Sila	12°08′25.3″	15°03′25.6″
S138	white	Geida Sila	12°08′25.3″	15°03′25.6″
S139	white	Geida Sila	12°08′25.3″	15°03′25.6″
S140	beige	Geida Sila	12°08′25.3″	15°03′25.6″
STh ^a^	black	Market	-	-

^a^ STh (*S. indicum* from Thailand) was purchased on the market.

**Table 2 genes-15-00711-t002:** Genome size estimates and sample data. Mean 2C value, standard deviation (SD), minimum (Min) and maximum (Max) 2C values in pg and mean 1Cx value in Mbp obtained for each accession. Group membership according to agro-morphological analyses (AgroM) [22] and genetic diversity analyses (Genetic) [21] are indicated for each Niger accession. n: number of individuals assessed.

Accession	Genome Size (pg)				Mean 1Cx Value in Mbp ^b^	Sample Data	
	n	Mean 2C Value	S.D.	Min–Max		AgroM ^c^	Genetic ^d^
S1	5	0.87	0.008	0.86–0.88	426	AgroM2	Gr2
S2	5	0.87	0.012	0.86–0.89	425	AgroM1	Gr1
S3	5	0.85	0.011	0.83–0.86	414	AgroM1	Gr1
S4	5	0.84	0.008	0.83–0.85	410	AgroM3	Gr3
S7	5	0.87	0.013	0.86–0.89	426	AgroM3	Gr3
S8	5	0.88	0.020	0.86–0.91	430	AgroM3	Gr1
S11	5	0.84	0.025	0.82–0.88	411	AgroM3	Gr3
S12	5	0.90	0.031	0.87–0.94	440	AgroM1	Gr1
S13	5	0.91	0.008	0.9–0.92	446	AgroM3	Gr3
S14a	5	0.88	0.011	0.86–0.89	428	AgroM2	Gr1
S15	5	0.80	0.007	0.79–0.81	391	AgroM2	Gr1
S16	5	0.87	0.018	0.86–0.90	426	AgroM2	Gr2
S17	5	0.81	0.005	0.80–0.81	394	AgroM2	Gr2
S22b	5	0.84	0.018	0.82–0.86	412	AgroM3	Gr3
S25	5	0.85	0.026	0.82–0.88	417	AgroM3	Gr3
S26	5	0.83	0.019	0.80–0.85	406	AgroM3	Gr3
S27	5	0.86	0.034	0.82–0.91	422	AgroM1	Gr1
S28	5	0.88	0.018	0.87–0.91	431	AgroM1	Gr1
S35	5	0.86	0.029	0.82–0.90	422	AgroM3	Gr3
S37	5	0.82	0.011	0.80–0.83	399	AgroM3	Gr3
S41	5	0.85	0.014	0.84–0.87	416	AgroM3	Gr1
S42	5	0.82	0.011	0.80–0.83	400	AgroM2	Gr2
S44	5	0.84	0.013	0.83–0.86	412	AgroM3	Gr3
S46	5	0.85	0.011	0.84–0.87	418	AgroM3	Gr3
S50	5	0.82	0.032	0.78–0.86	400	AgroM1	Gr1
S52	5	0.86	0.017	0.83–0.87	419	AgroM1	Gr1
S53	5	0.93	0.019	0.90–0.95	456	AgroM1	Gr2
S54	10	0.87	0.019	0.85–0.90	426	AgroM3	Gr3
S57	5	0.80	0.005	0.80–0.81	393	AgroM3	Gr3
S60	5	0.86	0.016	0.84–0.88	420	AgroM3	Gr2
S63	5	0.84	0.011	0.83–0.86	413	AgroM2	Gr1
S65	5	0.90	0.011	0.88–0.91	439	AgroM1	Gr3
S66	5	0.80	0.008	0.79–0.81	390	AgroM3	Gr2
S67	5	0.83	0.015	0.81–0.85	408	AgroM2	Gr1
S68	5	0.80	0.010	0.79–0.81	391	AgroM1	Gr3
S71	5	0.86	0.043	0.82–0.93	422	AgroM3	Gr3
S72	10	0.83	0.016	0.81–0.86	406	AgroM3	Gr2
S75	6	0.83	0.019	0.80–0.85	406	AgroM3	Gr3
S76	5	0.82	0.007	0.81–0.83	401	AgroM2	Gr2
S79	7	0.85	0.016	0.83–0.87	414	AgroM1	Gr1
S81	7	0.84	0.017	0.81–0.86	411	AgroM3	Gr3
S82	5	0.82	0.018	0.80–0.84	402	AgroM2	Gr2
S83	5	0.84	0.015	0.82–0.86	410	AgroM3	Gr3
S84	5	0.83	0.018	0.81–0.85	405	AgroM1	Gr1
S85	5	0.87	0.019	0.85–0.90	427	AgroM1	Gr1
S86	5	0.81	0.033	0.77–0.86	395	AgroM2	Gr2
S91	6	0.80	0.022	0.77–0.83	390	AgroM1	Gr1
S92	5	0.87	0.016	0.84–0.88	424	AgroM3	Gr3
S93	5	0.81	0.009	0.81–0.83	398	AgroM3	Gr3
S94	7	0.85	0.026	0.82–0.89	417	AgroM3	Gr3
S95	5	0.87	0.029	0.85–0.92	426	AgroM3	Gr3
S96	5	0.78	0.008	0.77–0.79	382	AgroM2	Gr2
S99	5	0.84	0.013	0.83–0.86	412	AgroM1	Gr1
S100	5	0.84	0.013	0.82–0.85	409	AgroM1	Gr1
S101	5	0.88	0.022	0.85–0.91	429	AgroM2	Gr2
S104	12	0.83	0.027	0.77–0.86	403	AgroM2	Gr2
S110	5	0.87	0.022	0.85–0.90	426	AgroM1	Gr1
S111	5	0.87	0.018	0.85–0.89	427	AgroM3	Gr3
S116	5	0.90	0.019	0.89–0.93	442	AgroM3	Gr3
S118	5	0.84	0.019	0.81–0.85	409	AgroM3	Gr2
S120	5	0.85	0.036	0.79–0.88	417	AgroM3	Gr3
S122	5	0.85	0.019	0.82–0.87	417	AgroM2	Gr2
S123	5	0.82	0.007	0.81–0.83	401	AgroM2	Gr3
S125	5	0.92	0.026	0.89–0.95	451	AgroM2	Gr2
S126	5	0.88	0.013	0.86–0.89	431	AgroM3	Gr3
S130	5	0.89	0.021	0.86–0.91	435	AgroM3	Gr3
S132	5	0.95	0.031	0.92–1.00	464	AgroM3	Gr3
S133	5	0.85	0.018	0.83–0.87	414	AgroM1	Gr1
S134	5	0.86	0.022	0.83–0.88	422	AgroM2	Gr2
S135	5	0.86	0.032	0.84–0.92	422	AgroM3	Gr3
S136	6	0.86	0.017	0.84–0.89	422	AgroM3	Gr3
S137	5	0.89	0.021	0.87–0.92	435	AgroM3	Gr3
S138	10	0.87	0.013	0.85–0.89	425	AgroM2	Gr2
S139	5	0.90	0.033	0.86–0.93	440	AgroM3	Gr3
S140	5	0.90	0.033	0.85–0.94	441	AgroM3	Gr3
STh ^a^	8	0.73	0.01	0.72–0.74	356	-	-

^a^ STh (*S. indicum* from Thailand) was purchased on the market; ^b^ 1 pg = 978 Mbp [47]; ^c^ AgroM—agr-morphological groups; ^d^ Genetic (Gr)—genetic groups.

**Table 3 genes-15-00711-t003:** Kruskall–Wallis tests performed on the genome size scored data from all accessions studied (All accessions), among the genetic and agro-morphological groups (Among groups) and within each Niger genetic (Gr1, Gr2 and Gr 3) or agro-morphological (AgroM1, AgroM2 and AgroM3) group.

		d.f.	*H*	*P* ^1^
All accessions	+STh	75	318.45	<2.2 × 10^−16^ ***
	−STh	74	306.65	
Genetic Groups	Among Groups (+STh)	3	35.1	1.161 × 10^−7^ ***
	Among Groups (−STh)	2	11.92	0.003 **
	Gr1	20	74.89	2.8 × 10^−8^ ***
	Gr2	18	89.68	1.7 × 10^−11^ ***
	Gr3	34	133.6	9.6 × 10^−14^ ***
Agro-morphological Groups	Among Groups (+STh)	3	35.78	8.3 × 10^−8^ ***
	Among Groups (−STh)	2	12.62	0.002 **
	AgroM1	17	70.77	1.6 × 10^−8^ ***
	AgroM2	18	87.08	7.31 × 10^−11^ ***
	AgroM3	37	141.99	3.2 × 10^−14^ ***

^1^ Adjusted *p*-value with Holm’s correction. Statistical significance was assigned at *p* < 0.01 (**) and *p* < 0.001 (***); ns: non-significant.

**Table 4 genes-15-00711-t004:** Spearman correlation analyses between 2C DNA values from the current study and data for different variables from previous work [22,23,24]. *r*: coefficient of correlation.

Variable	*r*	*p* ^1^
Branching	−0.05	0.64 ^ns^
Flowering Time	−0.27	0.02 *
Height	−0.045	0.70 ^ns^
Latitude	0.0002	0.99 ^ns^
Longitude	−0.06	0.59 ^ns^
Fatty acid content	−0.06	0.67 ^ns^
Linoleic acid content	−0.15	0.27 ^ns^
Oleic acid content	0.22	0.12 ^ns^
Seed maturity	−0.06	0.59 ^ns^
Yield	0.05	0.66 ^ns^

^1^ Adjusted *p*-value with Holm’s correction. Statistical significance was assigned at *p* < 0.05 (*); ns: non-significant.

**Table 5 genes-15-00711-t005:** Linear regression model.

		d.f.	*R Squared* ^1^	*F*	*P* ^2^
All accessions ^3^		1;73	0.045	4.517	0.04 *
Genetic Groups	Gr1	1;19	0.135	4.124	0.06 ^ns^
	Gr2	1;17	−0.056	0.048	0.83 ^ns^
	Gr3	1;33	0.027	1.945	0.17 ^ns^
Agro-morphological Groups	AgroM1	1;16	0.019	1.336	0.26 ^ns^
	AgroM2	1;17	−0.050	0.140	0.71 ^ns^
	AgroM3	1;36	0.026	1.969	0.17 ^ns^

^1^ Adjusted R-squared. ^2^ Adjusted *p*-value with Holm’s correction. ^3^ All Niger accessions. Statistical significance was assigned at *p* < 0.05 (*); ns: non-significant.

## Data Availability

The original contributions presented in the study are included in the article, further inquiries can be directed to the corresponding author.

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
