# Peer review of "Genome Size Variation in *Sesamum indicum* L. Germplasm from Niger"

_genes, 2024, doi:10.3390/genes15060711_

Round 1
Reviewer 1 Report
Comments and Suggestions for Authors
The ms reoprted the GS of different accessions based on flow cytometry and analyzed the coorelation between GS and phenotypes, it is a good topic, but I have several questions:
1,the introdcution part should be improved to declare the importance of the present study.
2, since the author pointed out the plant materials of 75 accessions were selected from 140 entires. they should tell how did they select these 75 acc.. and the accession from Thailand should be mentioned in the plant material part.
3, L135-136, the information was in the plant materials part, so it should be deleted from the result part.
4,L188, how did the author considered it was a significant level as the R value is only 0.27. and I don't think there is much credibility in the correlations between GS and flower time within species.
5, I would like suggest the authors present some information of the GS by Mb based on the present results in the disscution part, so it is easier for readers to understand, since this is the age of genome sequencing.
Comments on the Quality of English LanguageThe English writing of the whole article shoule be improved and should be simplifed.
Author Response
We would like to thank you for your valuable comments, which helped us to further improve our manuscript. All changes are indicated in red in the manuscript. We have also provided the text for the language revision (English improvement are not in red).
Here are our responses to your questions :
The ms reported the GS of different accessions based on flow cytometry and analyzed the correlation between GS and phenotypes, it is a good topic, but I have several questions:
1, the introdcution part should be improved to declare the importance of the present study.
Response: We improved the introduction and modified the last paragraph of the introduction to emphasise the importance of this study (L68-L71).
2, since the author pointed out the plant materials of 75 accessions were selected from 140 entires. they should tell how did they select these 75 acc. and the accession from Thailand should be mentioned in the plant material part.
Response: We added information about how the accessions were selected:
L77-L80: “The seventy-five accessions in the Niger germplasm were selected on the basis of their membership of previously identified genetic and/or agro-morphological groups and their geographical distribution (Figure 1, Table 1). Most of them have also been biochemically characterised in a previous study (Zangui et al. 2019)”.
L80: L81: The Tai sesame accession was mentioned in the paragraph on plant material: “A Tai sesame accession (STh) purchased on the market was included in the studied sample”.
3, L135-136, the information was in the plant materials part, so it should be deleted from the result part.
Response: The sentence has been deleted.
4, L188, how did the author considered it was a significant level as the R value is only 0.27. and I don't think there is much credibility in the correlations between GS and flower time within species.
Response: We appreciate your attention to detail regarding the correlation coefficient in our study. However, it is important to note that the significance of the correlation is determined by the p-value, which in this case is P = 0.02 < 0.05, indicating statistical significance. While the correlation coefficient (r) of -0.27 does indeed indicate a moderate correlation, the significance remains robust. The following sentence has been adapted for clarity:
L195-L196: “Although Spearman’s correlation coefficient was moderate (r= -0.27), it was still statistically significant (P = 0.02).”
5, I would like suggest the authors present some information of the GS by Mb based on the present results in the disscution part, so it is easier for readers to understand, since this is the age of genome sequencing.
Response: Information on the mean C-value in Mbp is already given in Table 2 in the "Results" section. It has also been added to the “Abstract” and “Discussion” sections.
Comments on the Quality of English Language
The English writing of the whole article should be improved and should be simplifed.
Response: The language revision is done.
Reviewer 2 Report
Comments and Suggestions for Authors
Dear Editor and colleagues,
I have read with interest the manuscript “Genome size variation in Sesamum indicum L. germplasm from Niger” submitted in genes-mdpi.
This is a work describing the genomic characterization of African sesame accessions based on C-values determination. Since there aren’t many papers dealing with genome size estimation in sesame, I believe that this work offers significant insights into the literature. The work is well designed and written
Still, some caveats need clarification.
· In FCM analyses it is generally advised to use a standard where 2C and 4C values among sample and standard do not coincide. Tomato has been chosen (cv. Stupické polní tyčkové rané is the ‘golden standard’ having 2C=1.96 pg), where Raphanus sativus cv. Saxa (2C=1.11 pg) might be a better choice.
· There are no reports of CVs in tables or figures. In general, histograms having a CV<5% must be acquired to get reliable C-value readings.
· In figures, 2C and 4C populations must be indicated.
· Please add boxplots with C-values according to AgroMc /Genetic types to summarize C-values and help readers’ understanding.
· Table 2 and Table A1 lack statistically significant values (Anova groups)
And a suggestion:
· Since there are genetic groups based on AFLPs, a character evolution for C-values would greatly enhance information (via Mesquite or similar software). please see: Charalambous, I.; Ioannou, N.; Kyratzis, A.C.; Kourtellarides, D.; Hagidimitriou, M.; Nikoloudakis, N. Genome Size Variation across a Cypriot Fabeae Tribe Germplasm Collection. Plants 2023, 12, 1469. https://doi.org/10.3390/plants12071469
Based on the above I recommend a major revision
Comments on the Quality of English Language-
Author Response
We would like to thank you for your valuable comments, which helped us to further improve our manuscript. All changes are indicated in red in the manuscript.
Here are our responses to your questions :
I have read with interest the manuscript “Genome size variation in Sesamum indicum L. germplasm from Niger” submitted in genes-mdpi.
This is a work describing the genomic characterization of African sesame accessions based on C-values determination. Since there aren’t many papers dealing with genome size estimation in sesame, I believe that this work offers significant insights into the literature. The work is well designed and written
Still, some caveats need clarification.
- In FCM analyses it is generally advised to use a standard where 2C and 4C values among sample and standard do not coincide. Tomato has been chosen (cv. Stupické polní tyčkové rané is the ‘golden standard’ having 2C=1.96 pg), where Raphanus sativus cv. Saxa (2C=1.11 pg) might be a better choice.
Response: Thank you for the advice.
However, we believe that the use of Solanum lycopersicum L. (‘Montfavet 63-5’, 2C = 1.99 pg) as internal standard was the most appropriate choice. In almost every histogram, there was no overlap between peaks. In addition, the 4C of sesame is very poorly represented and only overlaps with 2C of standard in the case of a few accessions with the largest GS. However, if we were to use Raphanus sativus as standard, the 2C peak of certain sesame accessions could be too close to 2C of Raphanus sativus, which could be even more disturbing.
- There are no reports of CVs in tables or figures. In general, histograms having a CV<5% must be acquired to get reliable C-value readings.
Response: In the case of measurements on sesame cotyledons, despite the good peaks obtained, the CV was ranging from 3.45 to 8.07 % while the CV for the standard leaf was still around 3%. For the moment, we have no explanation for this difference in CV between sesame and tomato other than the fact that they are two slightly different tissues. If necessary, we could add in Table 2 or in supplementary data (Table A1) the mean CV for each accession.
In figures, 2C and 4C populations must be indicated.
Response: Done.
- Please add boxplots with C-values according to AgroMc /Genetic types to summarize C-values and help readers’ understanding.
Response: Done. We added the figure of boxplots to the "Results" section (Figure 3) on page 9.
- Table 2 and Table A1 lack statistically significant values (Anova groups)
Response: In the supplementary data we have added Table A3, which presents the results of pairwise comparisons using Dunn's test for the 58 pairs of accessions showing a significant difference in GS (P<0.05). All other pairs of accessions showing no significant difference in GS (P>0.05) are not presented.
And a suggestion:
- Since there are genetic groups based on AFLPs, a character evolution for C-values would greatly enhance information (via Mesquite or similar software). please see: Charalambous, I.; Ioannou, N.; Kyratzis, A.C.; Kourtellarides, D.; Hagidimitriou, M.; Nikoloudakis, N. Genome Size Variation across a Cypriot Fabeae Tribe Germplasm Collection. Plants 2023, 12, 1469. https://doi.org/10.3390/plants12071469
Thank you very much for this interesting suggestion, which we will apply in the future work that we are already planning to carry out on a larger number of representatives of the Sesamum genus.
Round 2
Reviewer 1 Report
Comments and Suggestions for Authors no comments Comments on the Quality of English Languagethe English language is ok now
Reviewer 2 Report
Comments and Suggestions for Authors
the authors have provided adequate data and the revised manuscript has been improved. i believe the MS can be accepted
Comments on the Quality of English Language-